# A Cross-Sectional Pilot Study on Food Intake Patterns Identified from Very Short FFQ and Metabolic Factors Including Liver Function in Healthy Japanese Adults

**DOI:** 10.3390/nu14122442

**Published:** 2022-06-13

**Authors:** Miya Uchiyama, Chizuko Maruyama, Ariko Umezawa, Noriko Kameyama, Aisa Sato, Kanako Kamoshita, Seina Komine, Sayaka Hasegawa

**Affiliations:** 1Division of Food and Nutrition, Graduate School of Human Sciences and Design, Japan Women’s University, 2-8-1 Mejiro-dai, Bunkyo-ku, Tokyo 112-8681, Japan; m1622302om@gr.jwu.ac.jp (M.U.); umezawaa@fc.jwu.ac.jp (A.U.); m1403022sa@ug.jwu.ac.jp (A.S.); 2Department of Food and Nutrition, Faculty of Human Sciences and Design, Japan Women’s University, 2-8-1 Mejiro-dai, Bunkyo-ku, Tokyo 112-8681, Japan; kameyaman@fc.jwu.ac.jp (N.K.); m1203019kk@ug.jwu.ac.jp (K.K.); m1303019ks@ug.jwu.ac.jp (S.K.); m1303041hs@ug.jwu.ac.jp (S.H.)

**Keywords:** dietary pattern, Japanese, Westernized, non-alcoholic fatty liver disease, BMI, liver function, food frequency questionnaire, nutrition education

## Abstract

Non-alcoholic fatty liver disease is a growing health problem, and rapid diet assessment is required for personal nutrition education. This pilot study aimed to clarify associations between current food intake patterns identified from the short food frequency questionnaire (FFQ) and metabolic parameters, including liver function. We conducted a cross-sectional study on Japanese non-alcoholic residents of Tokyo and surrounding districts, 20 to 49 years of age. Anthropometric measurements, fasting blood samples, three-day dietary records, and FFQ with 21 items were collected. In all 198 participants, the proportions with obesity were 21% in men and 6% in women. Hypertriglyceridemia was significant only in men, affecting 26%. The traditional Japanese (TJ) pattern (greater intakes of green and yellow vegetables, other vegetables, seaweed/mushrooms/konjac, dairy, fruits, fish, salty, and soybeans/soy products) and the Westernized pattern (greater intakes of saturated-fat-rich foods, oily, egg/fish-eggs/liver, and sweets) were identified. The TJ pattern score showed an inverse relationship with body mass index, triglyceride, alkaline-phosphatase, leucine-aminopeptidase, and fatty liver index. The TJ pattern identified from the short FFQ was suggested to be associated with body fat storage. Further large-scale studies are needed to clarify the associations between this dietary pattern and metabolic parameters, including liver function.

## 1. Introduction

Non-alcoholic fatty liver disease (NAFLD) and non-alcoholic steatohepatitis (NASH) are increasing health problems, with rising rates of obesity globally. Preventing NAFLD and NASH is crucial in Japan where obesity with a body mass index (BMI) exceeding 25 had increased from 16% in the 1970s to 30% in men in their 30s and 40% in those in their 40s and 50s in 2019 [1].

Despite the Japanese dietary pattern in the 1970s through 1980s having been regarded as healthy [2,3,4,5], Japanese lifestyle and social environmental factors concerning food suppliers have changed markedly with concurrent Westernization: increased intake of animal foods, such as red meat, meat products, and dairy, and decreased intakes of rice, fish, and vegetables. Furthermore, diversification has increased the frequencies of eating out or take-out meals and ready-to-eat foods/meals and decreased chances to eat meals together with family members. These changes have caused different food intakes with unbalanced nutrient intakes, making healthy eating difficult for the younger generation [6], especially those living in urban areas of Japan [7]. Body weight reduction by calorie intake restriction and exclusion of added fructose has been recommended as a diet therapy for NAFLD [8,9,10,11]. Furthermore, antioxidant intake is considered to be beneficial for preventing the progression of NASH, and the Mediterranean diet is recommended as a potential therapy for patients with NAFLD in European countries [12,13]. A few Japanese cross-sectional studies have examined dietary patterns related to abdominal obesity or NAFLD prevalence, and the traditional Japanese dietary pattern is recommended for management of NAFLD [14]. When assessing actual targets relevant to participants’ habitual diets and providing advice aimed at ameliorating health problems quickly in the clinical practice of personal nutrition education, time-consuming diet surveys are not practical. We have thus developed a very short food frequency questionnaire (FFQ) as a method of implementing clinical nutrition education targeting atherosclerotic disease risk factors, including metabolic syndrome [15].

This pilot study aimed to clarify the associations between current dietary patterns identified from the short FFQ and metabolic parameters, including liver function, in young non-alcoholic Japanese adults who had been brought up in the highly Westernized dietary environment of modern Japan.

## 2. Materials and Methods

### 2.1. Study Design and Participants

This was a cross-sectional study conducted according to the guidelines of the Declaration of Helsinki, and all procedures were approved by the Ethics Committee for Experimental Research Involving Human Subjects of Japan Women’s University (No.265). We obtained written informed consent from all subjects prior to enrollment. The clinical trial registration number is UMIN000024195.

Japanese residents of Tokyo and surrounding districts, 20 to 49 years of age, were recruited to participate in this study by poster, leaflet, and e-mail from September 2016 to October 2018. Patients taking medications, those who were pregnant, nursing women, those engaged in high-intensity exercise, and individuals habitually consuming dietary supplements or functional foods were excluded.

### 2.2. Anthropometrics, Blood Pressure, and Biochemical Measurements

Anthropometric measurements and fasting blood collection were conducted at Japan Women’s University in the morning following an at least 12 h fast. Body height and weight were measured, and BMI was calculated as weight (kg) divided by the square of height (m). A BMI of 22 kg/m^2^, which has been proposed as the ideal body for individuals 30–59 years old, who have the lowest morbidity in the Japanese population, was regarded as corresponding to an ideal body weight (IBW) for convenience in making simple estimations [16,17]. Umbilical circumference was determined with a tape measurer during the late exhalation phase in the standing position. Blood pressure was measured using an automatic blood pressure manometer with participants in a seated position.

Serum samples were obtained, and the following concentrations were measured: total cholesterol (enzymatic method), low-density lipoprotein cholesterol (LDL-C) (direct enzymatic method), high-density lipoprotein cholesterol (HDL-C) (direct enzymatic method), triglyceride (TG) (enzymatic method), aspartate aminotransferase (AST) (JSCC consensus method), alanine aminotransferase (ALT) (JSCC consensus method), alkaline phosphatase (ALP) (JSCC consensus method), leucine aminopeptidase (LAP) (colorimetric method), γ-glutamyl transpeptidase (γ-GT) (JSCC consensus method), total bilirubin, and direct bilirubin (colorimetric chemical oxidation method) at the Laboratory of BML Inc., Tokyo, Japan. The indirect bilirubin concentration was calculated from the total bilirubin minus direct bilirubin concentration. The “fatty liver index” (FLI) was calculated as: FLI = (e 0.953 × log (TG) + 0.139 × BMI + 0.718 × log (γ-GT) + 0.053 × waist circumference—15.745)/(1 + e 0.953 × log (TG) + 0.139 × BMI + 0.718 × log (γ-GT) + 0.053 × waist circumference—15.745) × 100, and a FLI ≥ 60 was taken to indicate the presence of hepatic steatosis, while FLI < 30 ruled it out [18].

### 2.3. Food Frequency Questionnaire

The very short food intake frequency questionnaire named “Plus1Minus1”, designed for assessment and evaluation in nutrition education practice by Maruyama et al., was used. The “Plus1Minus1” consists of 21 food items (unrefined cereals, fish, soybeans/soy products, green and yellow vegetables, other vegetables, seaweed/mushrooms/konjac, fruits, milk/cheese, yoghurt/probiotic drinks, confections/sweets, sugar-sweetened beverages, fatty meat/fatty poultry, bacon/ham/sausage, butter/margarine, mayonnaise/oil-based dressings, eggs/fish-eggs/liver, fried foods, salty foods (tsukemono: Japanese pickles, tsukudani: foods boiled in sweetened soy sauce, fishery paste products, and salted seafoods), soup, and alcoholic beverages [15]. For most food groups, the eating frequencies were each scored as follows: rarely: (1), 1 to 2 times/week: (2), 3 to 4 times/week: (3), 5 to 6 times/week: (4), over 7 times/week: (5). For green and yellow vegetables and other vegetables, the scores were as follows: less than 2 times/week: (1), 3 to 4 times/week: (2), 5 to 6 times/week: (3), 7 to 10 times/week: (4), 11 or more times/week: (5). Soup intake was scored as less than twice/week: (1), 3 to 4 times/week: (2), 5 to 6 times/week: (3), 7 to 10 times/week: (4), 11 or more times/week: (5). The scores for milk/cheese and yoghurt/probiotic drinks, confections/sweets and sugar-sweetened beverages, mayonnaise/oil-based dressings and fried foods, salty foods, and soup were each summed and divided by two, then rounded to the nearest integer from 1 to 5, as follows: rarely: (1), less than 4 times/week: (2), 4 to 8 times/week: (3), 8 to 12 times/week: (4), 12 or more times/week: (5). The scores for fatty meat/fatty poultry, bacon/ham/sausage, and butter/margarine were each summed and divided by three, then rounded to the nearest integer from 1 to 5, as follows: less than twice/week: (1), 2 to 7 times/week: (2), 8 to 13 times/week: (3), 14 to 18 times/week: (4), 19 or more times/week: (5). These combined food groups were designated dairy, sweets, oily, salty, and saturated-fat-rich foods, as appropriate.

### 2.4. Dietary Records and Physical Activity

Participants were asked to keep three-day (two weekdays and one weekend day) dietary records during the week just before the examinations. They were asked to weigh and record all foods and beverages consumed on each day of recording. When weighing was difficult, e.g., dining-out or when eating take-out dishes on the go, the participants were instructed to take photos of their meals, as well as food and nutrient labels of the prepared foods they ate. Dietary records were confirmed and collected by a registered dietitian. Energy and nutrient intakes were calculated employing Excel-Eiyokun Version 8.0 software (Kenpakusha Co., Ltd., Tokyo, Japan) based on the “Standard Tables of Food Composition in Japan 2015”, seventh revised edition (Ministry Education, Culture, Sports, Science and Technology, Japan).

The volumes consumed were summarized into food groups in accordance with nutrient content similarities: cereals and potatoes, confections, sugar and jam, sugar-sweetened beverages, meat/poultry, meat products, milk and dairy products, eggs/fish-egg/liver, soybean/soy products, green and yellow vegetables, other vegetables, seaweed/mushrooms/konjac, fruit, butter and margarine, oily seasonings (vegetable oils, mayonnaise, oil-based dressings and roux), salted products (pickles, tsukudani, fishery paste products, and salted seafoods), non-oil seasonings (non-oil dressing, soy sauce, vinegar, mirin, Worcestershire sauce, miso, etc.), and alcoholic beverages. The alcohol intake values were calculated as pure ethanol amount in grams. For nutrients and food group intakes, the average intakes for three days were taken as the participants’ daily intakes. The food intake amount was adjusted by IBW in order to minimize differences due to actual body weights, as reflected by sex and body size.

Physical activity was determined by employing a questionnaire to calculate energy expenditure [19]. Participants were asked about the average time per day spent on each of the following activities: heavy physical work or strenuous exercise (assigning the value of 0 for “none,” 0.5 for “less than 1 h,” and 3 for “1 h and more”), sedentary activity (1.5 for “less than 3 h,” 5.5 for “3–8 h,” and 7.5 for “more than 8 h”), and walking or standing (0.5 for “less than 1 h,” 2 for “1–3 h,” and 8.5 for “more than 3 h”). Metabolic equivalent (MET) hours per day were estimated by multiplying the reported time spent on each activity per day by its assigned MET intensity [20]: heavy physical work or strenuous exercise (4.5), walking or standing (2.0), sedentary (1.5), and sleep and others (1.5).

### 2.5. Statistical Analysis

Statistical analyses were carried out using IBM SPSS Statistics (Version 26: IBM Japan Ltd. Tokyo, Japan). We used descriptive statistics with the means +/− standard deviation (SD) and median (interquartile range (IQR)), which are also presented for each continuous variable. Categorical variables are expressed as percentages. To compare differences between men and women, the unpaired-*t* test (parametric distribution) and the Mann–Whitney U-test (non-parametric distribution) were used. The Chi-squared test was used to determine the significance of differences in proportions of men and women.

Food intake patterns were identified using principal component analysis based on intakes of 13 food groups (unrefined cereals, fish, soybeans/soy products, green and yellow vegetables, other vegetables, seaweed/mushrooms/konjac, fruits, dairy, sweets, saturated-fat-rich food, eggs/fish-egg/liver, oily, salty) defined by the “Plus1Minus1”. We considered eigenvalues, the scree plot, and the interpretability of the factors to determine the number of factors to retain and identified two dietary factors. The dietary patterns were named according to the food items showing high loading values, i.e., larger than 0.4, in each dietary pattern. The factor scores for each dietary pattern in an individual were estimated by summing intakes of food items weighted by their factor loadings. We used these scores to rank participants according to each dietary pattern by dividing dietary pattern scores into tertiles. Therefore, each participant was grouped into these food intake pattern tertiles based on his/her scores, and we then evaluated the magnitude of the associations between each of these dietary patterns and food intakes, nutrient intakes, and biochemical parameters. Trend associations across the tertiles for each of the two food intake pattern scores were assessed by applying the Jonckheere–Terpstra test. A value of *p* < 0.05 was considered as indicating statistical significance.

## 3. Results

In total, 213 participants were initially enrolled in this study, of whom 14 were excluded due to being habitual drinkers whose ethanol intakes exceeded 30 g/day and 20 g/day, respectively, for men and women, and one who drank heavily on the previous evening of the survey. Thus, 198 (96 men and 102 women) subjects were included in all analyses. Study participants who worked full time accounted for 85.4% of men and 67.6% of women, and 10.8% of the women were housewives.

Most of the physical and biochemical parameters except HDL-C were higher in men than women (Table 1). The proportions with obesity, i.e., those whose BMI was 25 kg/m^2^ or higher, were 20 (20.8%) and 6 (5.9%), respectively, in men and women, and hypertriglyceridemia, i.e., TG of 150 mg/dL or higher, was seen only in men, affecting 24 (25.0%). The number of participants whose FLI exceeded 60 was 13 (6.6%), while 169 (85.3%) had FLI below 30.

Distributions of food intake frequencies according to the “Plus1Minus1” FFQ are shown in Figure 1. Soybean/soy products, seaweed/mushroom/konjac, and fish were consumed by 25.8%, 20.2%, and 5.0% of the participants, respectively, more than 5 times per week. Food groups accounting for a higher percentage in the “rarely” intake category were unrefined cereals in 71.2% of the participants, followed by fruits, green and yellow vegetables, saturated-fat-rich food, and fish in 33.8%, 15.7%, 15.7%, and 14.1%, respectively.

Two food intake patterns were identified by principal component analysis that corresponded to the “Traditional Japanese” pattern (greater intakes of green and yellow vegetables, other vegetables, seaweed/mushrooms/konjac, dairy, fruits, fish, salty, and soybeans/soy products) and the “Westernized” pattern (greater intakes of saturated-fat-rich foods, oily, eggs/fish-eggs/liver, and sweets). These two food intake patterns accounted for 28.4% and 13.9%, respectively, and explained 42.3% of total variance in food intake (Table 2).

The median daily food intake volumes in grams per IBW according to the tertiles of the two food-intake pattern score groups are shown in Table 3. There were more men than women in the lowest and fewer in the highest tertile score group of the traditional Japanese food intake pattern (*p* = 0.007). The traditional Japanese food intake pattern was positively associated with green and yellow vegetables, other vegetables, fruit, milk, and dairy products (*p* for trend <0.001), fish, soybeans/soy products, seaweed/mushroom/konjac, non-oil seasonings (*p* for trend <0.01), and eggs/fish-eggs/liver (*p* for trend <0.05), while being inversely associated with sweetened beverages (*p* for trend <0.05). Salty products tended to be positively associated with the traditional Japanese food intake pattern (*p* for trend =0.073). The Westernized food intake pattern was positively associated with meat products (*p* for trend <0.001), butter and margarine, eggs/fish-eggs/liver (*p* for trend <0.01), and sweets (*p* for trend <0.05), while being inversely associated with green and yellow vegetables (*p* for trend <0.01) and soybeans/soy products (*p* for trend <0.05). In the third tertile score group of each traditional Japanese and Westernized food intake pattern, the ratio of the sum of meat/poultry, meat products, and egg/fish-eggs/liver intake volumes (Traditional Japanese: 2.13 g/IBW, Westernized: 2.33 g/IBW) and that of fish and soybean/soy products (Traditional Japanese:1.27 g/IBW, Westernized: 0.65 g/IBW) were 1.7 and 3.6, respectively. 

The median nutrient intake volumes in grams per IBW according to the tertiles of the two food-intake pattern score groups are presented in Table 4. The traditional Japanese food intake pattern showed strong positive associations with dietary fiber, energy percent derived from protein, potassium, β-carotene, vitamins C and D (*p* for trend <0.001), and the sum of eicosapentaenoic acid (EPA) and docosahexaenoic acid (DHA) (*p* for trend <0.01). The Westernized food intake pattern was positively associated with energy, energy percent derived from saturated fatty acids and polyunsaturated fatty acids, cholesterol (*p* for trend <0.01), and sodium (*p* for trend <0.05), while being negatively associated with dietary fiber (*p* for trend <0.001) and β-carotene (*p* for trend <0.01).

Anthropometric, blood pressure, and biochemical parameters according to the tertiles of the two food-intake pattern score groups are shown in Table 5. The traditional Japanese food intake pattern was inversely associated with BMI, TG, ALP, LAP (*p* for trend <0.05), and FLI (*p* for trend <0.01), while being positively associated with HDL-C (*p* for trend <0.05). None of these parameters showed associations with the Westernized food intake pattern.

## 4. Discussion

The very short FFQ “Plus1Minus1” defined two food intake patterns in our study participants, all of whom had been brought up in the highly Westernized dietary environment of modern Japan. The traditional Japanese food intake pattern features identified were greater intakes of vegetables, seaweed/mushrooms/konjac, soybeans/soy products, fruits, fish, dairy, and salty foods. These features of the traditional Japanese pattern are similar to the Mediterranean diet but also differ in that they include seaweed/mushroom/konjac and especially soybeans, as legumes, but are not characterized by consumption of extra-virgin olive oil and nuts [13].

Previous Japanese cohort studies, which obtained baseline data from 1980 to 1995, had already suggested that dietary patterns reducing animal foods and increasing fish, soybeans and soy products, vegetables, seaweed, mushrooms, and fruit as their main components are beneficial not only for preventing coronary heart disease but also as a cerebrovascular disease prevention strategy [21,22,23]. Furthermore, the “Japanese Food Score” comprising soy, fish, vegetables, fungi, seaweed, and fruit from priori-defined components showed a similar impact on cardiovascular disease risk reduction [24]. However, recent cross-sectional studies conducted after the 2000 baseline did not identify similar dietary patterns, which included fish, seaweed, and/or mushrooms [25,26,27,28,29], and the numbers of component foods were small. The consumption of these foods is decreasing in the younger generation in Japan [30,31], which might be reflected in the results of cross-sectional studies showing a lack of fish, seaweed, and/or mushrooms in dietary patterns [25,26,27,28,29], which suggests changing dietary patterns in the Japanese population. The reason for our traditional type of food intake pattern being composed of these food items might be that the characteristics of our participants included many who were quite healthy and comparatively lean, especially the women.

Two studies, including elderly participants, investigated the associations between dietary patterns and the incidence of NAFLD diagnosed by the presence of fatty liver on abdominal ultrasonography. Seaweed, vegetables, mushrooms, and soy products were common components of the dietary patterns, which were associated with a lower risk of NAFLD in these two studies [32,33].

To our knowledge, this is the first study to examine the associations among liver function parameters and food intake patterns in Japan. We found the traditional Japanese food intake pattern score to be inversely related to parameters reflecting fat accumulation, such as BMI, TG, ALP, LAP, and FLI. High levels of ALP and LAP without an increase in the bilirubin concentration might reflect lipid accumulation. Recently, ALP has attracted attention as an independent predictor of NAFLD [34,35,36]. As we did not measure ALP by the International Federation of Clinical Chemistry and Laboratory Medicine (IFCC) method, our data might include intestinal ALP. However, serum samples were collected after a 12 h fast, such that most of the measured ALP would have been of hepatic origin.

As the participants were healthy volunteers, we considered FLI to represent the presence of steatosis. Most of the participants had low FLI values, which ruled out fatty liver, as expected, though even the highest traditional Japanese food-intake pattern score group had lower FLI than the other lower score groups. Tien et al. reported a nutrient pattern rich in folate, carotene, vitamin C, fiber, iron, and potassium, containing an abundance of vegetables and fruits, to be associated with a lower prevalence of NAFLD diagnosed based on FLI ≥ 60 in a Japanese population residing in Tokushima Prefecture [37].

The foods comprising the traditional Japanese food intake pattern in this study would be expected to contribute to body fat reduction with higher intakes of dietary fiber, EPA and DHA, and antioxidant consumption associated with β-carotene, vitamin C, E, and D. Vegetables, seaweed/mushrooms/konjac, fruits, and soybeans/soy products are low in energy but fiber- and mineral-rich; fish is rich in EPA and DHA, fish and mushrooms are rich in niacin and vitamin D, and soybeans/soy products and fish can serve as main dishes rather than meats rich in saturated fatty acids. Dairy foods supply calcium, B vitamins, and vitamin D. Recent studies have indicated that vitamin D and niacin are involved in the modulation of metabolic and inflammatory pathways associated with the development of NAFLD [38,39].

The Japan Atherosclerotic Society recommended a dietary pattern designated “The Japan Diet”, featuring reduced intake of fat from meat and animal fats, such as beef tallow, lard, and butter, and increased combined consumptions of soy, fish, vegetables, seaweed, mushrooms, fruits, and unpolished grains, with low-salt cooking, to achieve clinical nutrition therapy [40]. We previously reported that nutrition education promoting the Japan Diet resulted in noticeable improvements in BMI, LDL-C, TG, and insulin levels, even with the inclusion of participants prescribed lipid-lowering drugs [41,42].

These results support the traditional Japanese food intake pattern being associated with body fat storage. However, sodium intakes did not differ among the traditional Japanese food-intake pattern score groups in our current study participants, such that the question of whether salty foods (tsukemono, soup, salted fish, etc.) should be included in the traditional Japanese food intake pattern remains to be answered, as these foods might increase the risk of high blood pressure. The traditional Korean dietary pattern characterized largely by the same foods, such as fermented, cruciferous, and green vegetables, fish, mushrooms, fermented soybeans, seaweed, and shellfish, was reported to have the association with an increased risk of NAFLD [43]. Further study on intake volumes, as well as cooking and seasoning methods, are needed to clarify the factors underlying these differences in observed results.

The other food intake pattern identified, i.e., the Westernized pattern, comprises greater intakes of saturated-fat-rich foods, oils, eggs/fish-eggs/liver, and sweets. This pattern would be expected to contribute to high energy consumption due to the saturated fatty acids, cholesterol, and fructose. Reductions in the consumptions of these foods have been recommended to prevent health problems related to insulin resistance, including NAFLD [44,45]. However, we detected no parameters significantly related to fat accumulation in the Westernized pattern group. Even the highest tertile in the Westernized pattern score group, meat/poultry and meat products, animal fats, such as butter and margarine, and eggs/fish-eggs/liver, were consumed at essentially the average volume eaten by the Japanese population [46]. In addition, fish, soybeans/soy products, vegetables, seaweed/mushrooms/konjac, fruits, and milk and dairy products were consumed in combination with foods featured in the Westernized food intake pattern. The prevalence of obesity was low in our study participants, such that the Westernized food intake pattern was not suggested to have a deleterious impact on body fat accumulation.

### Limitations

This study has limitations. First, many factors, such as lack of physical activity, genetic differences, and complex diseases interacting with non-dietary components, are important risks for NAFLD. However, we could not analyze the effects of these potentially confounding factors in the present small dataset. Second, as hepatic fat accumulation was not measured, we can draw no conclusions about the relationships between NAFLD and the identified dietary patterns. Third, as this was a pilot study, the number of participants was small, such that men and women could not be analyzed separately, and the fatty liver incidence is reportedly higher in men [47]. Fourth, there were very few obese participants, especially among the women. Fifth, the ALP measurements included intestinal ALP, as we used the JACC method. However, serum was obtained after a 12 h fast, such that most of the measured ALP would presumably have been of hepatic origin. Sixth, the FFQ “Plus1Minus1” used in this study was designed as a nutrition education tool and does not allow assessment of the quantity of food consumed.

## 5. Conclusions

The traditional Japanese dietary pattern featuring greater intakes of vegetables, seaweed/mushrooms/konjac, soybeans/soy products, fruits, fish, and dairy, was suggested to be associated with body fat storage, though our results are preliminary and must be interpreted cautiously. Further large-scale studies are needed to assess more precisely the associations between dietary patterns identified from the short FFQ and metabolic parameters, including liver function, in the current highly diverse Japanese eating environment.

## Figures and Tables

**Figure 1 nutrients-14-02442-f001:**
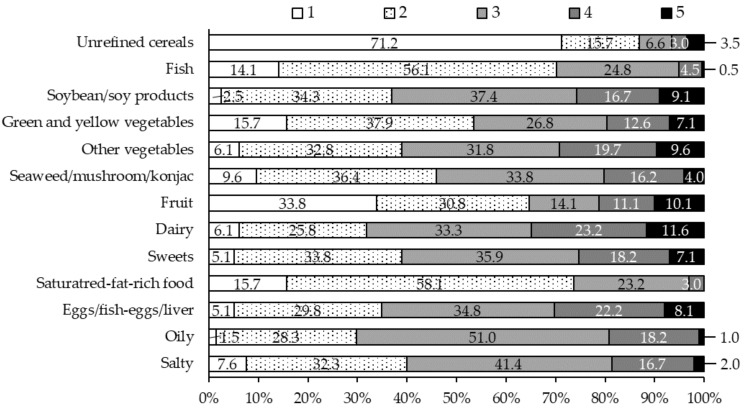
Distribution of food group intake frequency. Frequencies of unrefined cereals, fish, soybean/soy products, seaweed/mushroom/konjac, fruit, eggs/fish-eggs/liver were scored as follows: rarely: (1), 1 to 2 times/week: (2), 3 to 4 times/week: (3), 5 to 6 times/week: (4), over 7 times/week: (5). Frequencies of green and yellow vegetables and other vegetables were scored as follows: less than 2 times/week: (1), 3 to 4 times/week: (2), 5 to 6 times/week: (3), 7 to 10 times/week: (4), 11 or more times/week: (5). Each of the frequencies for dairy, sweety, oily, and salty were scored as follows: rarely: (1), less than 4 times/week: (2), 4 to 8 times/week: (3), 8 to 12 times/week: (4), 12 or more times/week: (5). The frequency of saturated-fat-rich foods was scored as follows: less than twice/week: (1), 2 to 7 times/week: (2), 8 to 13 times/week: (3), 14 to 18 times/week: (4), 19 or more times/week: (5).

**Table 1 nutrients-14-02442-t001:** Characteristics of participants.

Parameters	All (*n* = 198)	Men (*n* = 96)	Women (*n* = 102)	*p*
Age (years)	37 (28–44)	34 (27–43)	39 (30–46)	0.006
Height (cm)	164.5 (158.5–172.0)	172.0 (168.4–175.9)	158.6 (155.5–163.1)	<0.001
Weight (kg)	57.5 (50.8–66.4)	66.0 (60.6–73.0)	51.7 (47.9–55.8)	<0.001
Body mass index (kg/m^2^)	21.2 (19.8–23.0)	22.4 (20.8–24.1)	20.6 (19.4–21.8)	<0.001
Umbilical circumference (cm)	77.1 (72.5–82.7)	81.0 (74.9–85.0)	75.0 (71.0–79.1)	<0.001
SBP (mmHg)	111 (103–123)	121 (111–130)	106 (99–112)	<0.001
DBP (mmHg)	70 (64–77)	74 (67–84)	67 (62–74)	<0.001
Total cholesterol (mmol/L) †	5.05 (0.83)	5.02 (0.88)	5.07 (0.78)	0.690
LDL-C (mmol/L) †	2.86 (0.74)	2.97 (0.75)	2.77 (0.72)	0.037
HDL-C (mmol/L)	1.68 (1.42–1.95)	1.45 (1.27–1.68)	1.86 (1.68–2.07)	<0.001
Triglyceride (mmol/L)	0.69 (0.51–1.05)	0.89 (0.63–1.64)	0.59 (0.46–0.77)	<0.001
AST (U/L)	20 (18–24)	21 (18–26)	19 (16–21)	<0.001
ALT (U/L)	16 (12–23)	20 (16–28)	14 (11–17)	<0.001
ALP (U/L)	173 (146–210)	201 (167–234)	156 (131–179)	<0.001
γ-GT (U/L)	19 (14–30)	25 (18–38)	15 (12–24)	<0.001
LAP (U/L)	49 (44–55)	54 (49–61)	45 (41–49)	<0.001
Total bilirubin (μmol/L)	12 (10–15)	14 (10–15)	11 (9–14)	0.001
Direct bilirubin (μmol/L)	3 (3–5)	5 (3–5)	3 (3–3)	<0.001
Indirect bilirubin (μmol/L)	9 (7–10)	9 (7–10)	9 (7–10)	0.038
Fatty liver index	6.6 (3.4–17.4)	11.6 (6.1–31.2)	3.9 (2.4–8.7)	<0.001
Energy expenditure (kcal/day)	2218 (1986–2598)	2554 (2300–2973)	1989 (1875–2183)	<0.001

Values are presented as medians (IQR). †: Values are presented as means (SD). *p* values were calculated using the unpaired-*t* test (parametric distribution) and the Mann–Whitney U-test (non-parametric distribution) for differences between men and women. SBP: systolic blood pressure, DBP: diastolic blood pressure, LDL-C: low-density lipoprotein cholesterol, HDL-C: high-density lipoprotein cholesterol, AST: aspartate aminotransferase, ALT: alanine aminotransferase, ALP: alkaline phosphatase, γ-GT: γ-glutamyl transpeptidase, LAP: leucine aminopeptidase.

**Table 2 nutrients-14-02442-t002:** Factor loadings and explained variations for the dietary patterns identified by principal component analysis.

Food Groups	Traditional Japanese	Westernized
Unrefined cereals	0.376	−0.345
Fish	**0.549**	−0.293
Soybeans/soy products	**0.529**	−0.247
Green and yellow vegetables	**0.787**	−0.200
Other vegetables	**0.793**	−0.102
Seaweed/mushrooms/konjac	**0.748**	0.023
Fruits	**0.603**	−0.067
Dairy	**0.550**	0.282
Sweets	−0.024	**0.484**
Saturated-fat-rich food	0.207	**0.774**
Eggs/fish-egg/liver	0.382	**0.440**
Oily	0.094	**0.545**
Salty	**0.548**	0.293
Variance explained (%)	28.4	13.9

*n* = 198. Factor loadings >0.4 are shown in bold characters.

**Table 3 nutrients-14-02442-t003:** Food intakes according to tertiles of the two food intake pattern scores.

Food Groups	Traditional Japanese				Westernized			
	T1	T2	T3	*p* _trend_	T1	T2	T3	*p* _trend_
Men/Women (*n*)	41/25	32/34	23/43	0.007 #	33/33	31/35	32/34	0.941 #
Cereals and potatoes	6.60 (5.50–8.13)	6.75 (5.50–8.25)	6.65 (5.13–8.05)	0.768	6.50 (4.83–8.05)	6.90 (5.28–8.20)	6.75 (5.65–8.00)	0.412
Fish	0.10 (0.00–0.55)	0.29 (0.00–0.76)	0.47 (0.15–0.81)	0.001	0.46 (0.00–0.74)	0.39 (0.00–0.85)	0.25 (0.00–0.69)	0.233
Seafoods	0.13 (0.00–0.26)	0.08 (0.00–0.27)	0.16 (0.03–0.43)	0.070	0.21 (0.01–0.45)	0.09 (0.00–0.31)	0.11 (0.00–0.24)	0.075
Soybean/soy products	0.40 (0.18–1.13)	0.50 (0.18–0.93)	0.80 (0.30–1.40)	0.006	0.55 (0.28–1.63)	0.60 (0.28–1.13)	0.40 (0.10–0.90)	0.043
Green and yellow vegetables	0.70 (0.70–1.13)	1.10 (0.70–1.63)	1.50 (0.95–2.63)	<0.001	1.20 (0.60–2.23)	1.20 (0.80–2.20)	0.80 (0.40–1.40)	0.001
Other vegetables	1.65 (1.30-2.43)	1.90 (1.30–2.70)	2.75 (2.10–3.50)	<0.001	2.30 (1.38–3.20)	2.10 (1.45–2.90)	2.00 (1.30–2.80)	0.228
Seaweed/mushroom/konjac	0.20 (0.18–0.4)	0.25 (0.18–0.50)	0.40 (0.20–0.80)	0.001	0.30 (0.10–0.70)	0.30 (0.10–0.50)	0.30 (0.10–0.40)	0.392
Fruit	0.05 (0.00–0.4)	0.40 (0.00–1.80)	0.95 (0.30–1.83)	<0.001	0.60 (0.00–1.83)	0.30 (0.00–1.10)	0.30 (0.00–1.33)	0.265
Milk and dairy products	0.40 (0.30–1.53)	1.20 (0.30–2.95)	1.75 (1.08–2.63)	<0.001	0.95 (0.10–2.33)	1.25 (0.18–2.43)	1.30 (0.40–2.50)	0.127
Sweets	0.80 (0.28–1.43)	0.70 (0.28–1.23)	0.80 (0.30–1.23)	0.907	0.75 (0.20–1.33)	0.60 (0.10–1.03)	0.90 (0.50–1.60)	0.034
Sugar and jam	0.10 (0.10–0.20)	0.10 (0.10–0.10)	0.10 (0.10–0.20)	0.129	0.10 (0.10–0.20)	0.10 (0.00–0.10)	0.10 (0.10–0.20)	0.794
Sweetened beverages	0.95 (0.00–3.00)	0.25 (0.00–2.15)	0.00 (0.00–1.13)	0.032	0.00 (0.00–1.90)	0.00 (0.00–1.60)	1.00 (0.00–2.98)	0.063
Meat/poultry	1.41 (0.91–2.35)	1.58 (0.91–2.22)	1.41 (0.93–1.86)	0.306	1.43 (0.82–1.93)	1.59 (1.02–2.40)	1.43 (0.97–2.05)	0.350
Meat products	0.13 (0.00–0.35)	0.06 (0.00–0.26)	0.09 (0.00–0.33)	0.398	0.03 (0.00–0.24)	0.01 (0.03–0.22)	0.22 (0.06–0.42)	<0.001
Butter and margarine	0.03 (0.00–0.08)	0.03 (0.00–0.08)	0.03 (0.00–0.06)	0.329	0.02 (0.00–0.07)	0.02 (0.00–0.07)	0.04 (0.02–0.09)	0.004
Eggs/fish-egg/liver	0.54 (0.30–0.78)	0.56 (0.30–0.83)	0.63 (0.42–1.02)	0.011	0.49 (0.16–0.80)	0.58 (0.26–0.75)	0.68 (0.39–1.06)	0.002
Oily seasonings	0.34 (0.19–0.43)	0.37 (0.19–0.52)	0.28 (0.16–0.46)	0.583	0.28 (0.14–0.43)	0.41 (0.22–0.52)	0.34 (0.22–0.45)	0.305
Salty products	0.08 (0.00–0.26)	0.10 (0.00–0.31)	0.19 (0.03–0.53)	0.073	0.08 (0.00–0.39)	0.12 (0.05–0.45)	0.10 (0.01–0.32)	0.841
Non-oil seasonings	0.57 (0.39–0.70)	0.63 (0.39–0.86)	0.65 (0.53–0.92)	0.002	0.60 (0.41–0.85)	0.64 (0.46–0.87)	0.63 (0.47–0.74)	0.963

Values are presented as medians (IQR) in grams per ideal body weight. # *p* values were calculated using the Chi-squared test. *p*_trend_ values were calculated using the Jonckheere–Terpstra test.

**Table 4 nutrients-14-02442-t004:** Energy and nutrient intakes according to tertiles of the two food intake pattern scores.

Nutrients	Traditional Japanese				Westernized			
	T1	T2	T3	*p* _trend_	T1	T2	T3	*p* _trend_
Men/Women (***n***)	41/25	32/34	23/43	0.007 #	33/33	31/35	32/34	0.941 #
Energy (kcal/IBW)	30.2 (26.2–34.4)	32.2 (27.2–36.5)	32.8 (27.2–36.8)	0.053	30.1 (26.5–33.6)	31.6 (25.8–35.9)	33.3 (28.8–37.9)	0.008
Carbohydrate (%energy)	48.3 (44.6–53.0)	48.8 (45.2–53.4)	49.6 (45.2–53.2)	0.374	49.4 (45.1–55.2)	49.0 (45.3–53.0)	48.4 (44.8–52.1)	0.314
Dietary fiber (g/1000 kcal)	5.9 (4.7–6.5)	6.6 (5.3–7.9)	7.8 (5.3–9.6)	<0.001	7.5 (5.9–9.4)	6.9 (5.7–8.5)	6.0 (4.8–7.3)	<0.001
Protein (%energy)	14.1 (12.6–15.4)	14.0 (12.5–15.5)	15.5 (12.5–16.7)	<0.001	14.3 (13.2–16.1)	15.2 (13.3–16.5)	14.2 (12.6–15.4)	0.154
Fat (%energy)	33.3 (29.2–36.7)	32.9 (29.1–36.3)	32.2 (29.1–34.6)	0.243	31.4 (27.2–35.3)	32.7 (29.1–35.9)	33.7 (30.1–36.7)	0.018
Cholesterol (mg/IBW)	4.9 (3.3–6.4)	4.9 (3.6–6.6)	5.7 (3.6–7.4)	0.519	4.6 (3.6–5.9)	5.0 (4.2–6.7)	5.7 (4.3–7.2)	0.002
SFA (%energy)	10.2 (8.5–11.3)	9.8 (8.1–11.6)	9.7 (8.1–10.8)	0.075	9.4 (7.9–11.0)	9.4 (8.0–11.0)	10.2 (9.4–11.9)	0.001
MUFA (%energy)	12.8 (11.2–14.1)	12.5 (10.7–14.1)	11.6 (10.7–13.4)	0.445	11.4 (9.8–13.6)	12.4 (10.9–13.9)	13.0 (11.6–14.2)	0.522
PUFA (%energy)	6.4 (5.6–7.5)	6.5 (5.6–7.7)	6.3 (5.6–6.9)	0.004	6.2 (5.1–7.5)	6.8 (5.9–8.0)	6.1 (5.4–6.9)	0.005
n-6 PUFA (mg/IBW)	185 (145–219)	193 (152–231)	184 (152–221)	0.783	173 (128–221)	195 (160–232)	186 (154–222)	0.157
n-3 PUFA (mg/IBW)	33 (22–46)	33 (24–50)	37 (24–52)	0.036	35 (26–48)	37 (24–52)	33 (25–44)	0.632
EPA + DHA (mg/IBW)	3 (1–9)	5 (2–16)	10 (2–17)	0.001	7 (2–16)	7 (2–16)	4 (2–13)	0.415
β-carotene (μg/IBW)	30 (19–53)	46 (27–68)	60 (27–87)	<0.001	57 (29–87)	51 (28–73)	40 (24–56)	0.007
α-tocopherol (mg/IBW)	0.11 (0.09–0.12)	0.12 (0.10–0.15)	0.13 (0.10–0.16)	<0.001	0.11 (0.09–0.15)	0.13 (0.10–0.15)	0.11 (0.10–0.15)	0.432
Vitamin C (μg/IBW)	1.07 (0.76–1.61)	1.44 (0.96–2.01)	1.62 (0.96–2.10)	<0.001	1.53 (0.99–2.04)	1.34 (0.91–2.00)	1.25 (0.85–1.82)	0.172
Vitamin D(mg/IBW)	0.05 (0.03–0.09)	0.08 (0.04–0.12)	0.11 (0.04–0.17)	<0.001	0.08 (0.03–0.16)	0.08 (0.03–0.13)	0.07 (0.04–0.12)	0.588
Sodium (mg/IBW)	54.7 (47.4–63.2)	57.4 (46.9–68.9)	57.6 (50.0–65.0)	0.201	54.2 (43.7–63.1)	55.9 (48.6–68.1)	58.4 (52.4–66.5)	0.043
Potassium (mg/IBW)	34.6 (28.5–46.3)	40.7 (34.3–46.2)	43.3 (36.5–50.1)	<0.001	39.1 (34.9–49.8)	41.0 (33.4–49.0)	38.1 (30.5–46.1)	0.095

Values are presented as medians (IQR). # *p* values were calculated using the Chi-squared test. *p*_trend_ values were calculated using the Jonckheere–Terpstra test. IBW: ideal body weight, SFA: saturated fatty acids, MUFA: monounsaturated fatty acids, PUFA: polyunsaturated fatty acids, EPA: eicosapentaenoic acid, DHA: docosahexaenoic acid.

**Table 5 nutrients-14-02442-t005:** Anthropometric, blood pressure, and biochemical parameters according to tertiles of the two food intake pattern scores.

	Traditional Japanese				Westernized			
	T1	T2	T3	*p* _trend_	T1	T2	T3	*p* _trend_
Men/Women (*n*)	41/25	32/34	23/43	0.007 #	33/33	31/35	32/34	0.941 #
Height (cm)	167 (153–182)	166 (148–182)	163 (152–179)	0.013	165 (154–180)	165 (148–181)	166 (150–182)	0.684
Weight (kg)	62.3 (45.2–87.0)	60.3 (46.2–83.1)	56.9 (45.3–78.3)	0.002	59.4 (46.1–84)	59.9 (44.8–84.4)	60.2 (46.2–83.5)	0.639
Body mass index (kg/m^2^)	21.8 (20.3–23.6)	21.2 (19.7–23.7)	20.6 (19.5–22.4)	0.014	20.9 (19.8–22.4)	21.5 (19.7–23.3)	21.3 (20.3–23.6)	0.466
Umbilical circumference (cm)	78.6 (74.0–85.7)	78.4 (72.4–82.2)	75.2 (71.9–82.1)	0.053	76.4 (72.9–83)	76.3 (71.0–82.6)	78.2 (73.5–83.0)	0.599
SBP (mmHg)	112 (105–123)	112 (103–126)	108 (102–121)	0.100	109 (104–122)	112 (102–125)	112 (104–122)	0.470
DBP (mmHg)	73 (64–77)	70 (64–79)	70 (63–76)	0.224	69 (63–77)	70 (64–77)	72 (64–77)	0.218
Total cholesterol (mmol/L) †	5.17 (0.77)	4.94 (0.92)	5.04 (0.79)	0.249	5.01 (0.90)	5.09 (0.90)	5.05 (0.69)	0.489
LDL-C (mmol/L) †	2.97 (0.71)	2.80 (0.84)	2.82 (0.67)	0.225	2.86 (0.78)	2.83 (0.82)	2.89 (0.62)	0.840
HDL-C (mmol/L)	1.62 (1.34–1.99)	1.64 (1.29–1.93)	1.72 (1.57–1.95)	0.023	1.62 (1.29–1.93)	1.72 (1.42–2.02)	1.68 (1.47–1.89)	0.314
Triglyceride (mmol/L)	0.78 (0.55–1.31)	0.67 (0.52–1.00)	0.67 (0.46–0.93)	0.018	0.76 (0.53–1.03)	0.68 (0.50–1.05)	0.67 (0.50–1.06)	0.518
AST (U/L)	19(17–25)	20 (18–24)	20 (18–23)	0.817	19 (18–24)	20 (18–23)	19 (18–25)	0.833
ALT (U/L)	16 (12–25)	18 (14–24)	15 (12–20)	0.205	16 (12–23)	16 (13–25)	16 (12–22)	0.661
ALP (U/L)	188 (156–224)	175 (146–228)	163 (140–200)	0.031	172 (153–211)	189 (135–225)	168 (145–203)	0.659
γ-GT (U/L)	23 (14–34)	20 (15–33)	18 (14–26)	0.100	19 (14–31)	22 (15–35)	19 (14–30)	0.958
LAP (U/L)	51 (45–58)	49 (45–55)	47 (42–54)	0.019	48 (42–54)	49 (45–56)	50 (44–57)	0.190
Total bilirubin (μmol/L)	14 (10–15)	12 (9–15)	12 (10–14)	0.690	14 (10–15)	10 (9–15)	12 (10–14)	0.470
Direct bilirubin (μmol/L)	3 (3–5)	3 (3–5)	3 (3–5)	0.343	3 (3–5)	3 (3–5)	3 (3–5)	0.232
Indirect bilirubin (μmol/L)	9 (7–10)	8 (7–10)	9 (7–10)	0.922	9 (7–10)	8 (7–10)	9 (7–10)	0.619
Fatty liver index	9.4 (4.6–23.4)	6.8 (3.6–14.8)	5.7 (2.9–11.4)	0.003	6.6 (3.2–17.1)	6.6 (3.4–18.8)	6.7 (3.7–11.9)	0.937

Values are presented as medians (IQR). # *p* values were calculated using Chi-squared test. †: Values are presented as means (SD). *p*_trend_ values were calculated using the Jonckheere–Terpstra test. SBP: systolic blood pressure, DBP: diastolic blood pressure, LDL-C: low-density lipoprotein cholesterol, HDL-C: high-density lipoprotein cholesterol, AST: aspartate aminotransferase, ALT: alanine aminotransferase, ALP: alkaline phosphatase, γ-GT: γ-glutamyl transpeptidase, LAP: leucine aminopeptidase.

## Data Availability

Data sharing is not applicable to this article.

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
