# Peer review of "A Cross-Sectional Pilot Study on Food Intake Patterns Identified from Very Short FFQ and Metabolic Factors Including Liver Function in Healthy Japanese Adults"

_nutrients, 2022, doi:10.3390/nu14122442_

Round 1

Reviewer 1 Report

The paper is interesting. However, I have some suggestions to slightly enhance the manuscript.

Obesity, over-nutrition, dietary composition, and inactivity are important risk factors for NAFLD. But, these potential confounding factors were not discussed and controlled in the analysis. What were the difference of biochemical parameters, fatty liver index, energy, and nutrient intakes between two food intake patterns? Regarding the cross-sectional pilot study design, it could not confirm a causal relationship between dietary patterns and NAFLD. The application of very short FFQ for data collection may overestimate the participants’ energy intake, which is a great risk. And recall bias exists because of the self-reporting nature of the questionnaire.

A further explanation is needed.

 1. (Materials and Methods)

(Lines 74-75): A BMI of 22 kg/m2 was regarded as corresponding to the ideal body weight (IBW).

Why was BMI of 22kg/m2 regarded as corresponding to the ideal body weight? A further explanation is needed. Add the reference.

 2. (Lines 138-139): The food intake amount was adjusted by IBW, in order to minimize differences due to body size.

Why was the food intake amount adjusted by IBW instead of actual body weight? A further explanation is needed. Add the reference.

 3. (Results) There is no data for the following sentences.

(Lines 169-172): The proportions with obesity, i.e., whose BMI was 25 kg/m2 or higher, were 21% and 6%, respectively, in men and women, and hyper-triglyceridemia, i.e., TG of 150 mg/dL or higher was seen only in men, affecting 26.0%. The number of participants whose FLI exceeded 60 was 13 (6.6%), while 169 (85.3%) had FLI below 30.

(Lines 238-240): There were more men than women in the lowest and fewer in the highest tertile score group of the traditional Japanese food intake pattern (P for trend =0.007).

 4. Please revise this sentence - it is vague: Please clarify.

(Lines 247-249): In the 3rd tertile score group, the sum of the meat/poultry, meat products and egg/fish-eggs/liver intake volumes was almost double that of the fish and soybean/soy products.

 5. There are a few confused descriptions of data.

(Lines 151-152): ~~~ other vegetables, seaweed/mushrooms/konjac, fruits, dairy, sweetysaturated fat rich food, eggs/fish egg/liver, oily, salty) defined by the “Plus1Minus1”.

Insert ‘,’ between sweetysaturated fat rich food.

(Lines 243-244): ~~~ and eggs/fish-eggs/liver (P for trend <0.01), while being inversely associated with sweetened beverages (P for trend <0.05).

Correct P for trend of eggs/fish-eggs/liver as shown in Table 3.

Reviewer 2 Report

Thank you for the opportunity to read this interesting study about diet patterns (tradional Japanese and westernized) and associated physiological parameters. Overall this pilot study offered an interesting insight into the diet patterns and their associations with physiological parameters. Authors were clear about the limitations of their study in the discussion. However, considering the said limitations and the study design, authors’ conclusions appeared not always flow from the results. Further comments can be found below.

Abstract

Considering the observational and cross-sectional nature of this pilot-study, the conclusions appear not quite matching the actual results.

Introduction

Line 34: „…in Japan…“ – the sentence from here onwards is unclear. I believe that authors refer here to the different rates of obesity for middle aged (how is middle age defined in here?) men and women, but this should be written more clearly.

Line 38: “…westernization and diversification.” – please could the authors extrapolate briefly what the mean with this. Especially as lifestyles and culture are referred to.

Considering that the pilot study is cross-sectional, the study aim appears not matching the design. It is not possible to examine causal pathways, only associations. Based on the previous literature, did the authors have hypothesis they tested?

Methods

Line 69: Please could you be a bit more specific about what is meant with “health food products”.

Line 74: Please could you add a reference regarding the ideal body weight.

Line 138: Please could you add some information about how this was done. Also, as a reference to an earlier point raised, it is unclear why ideal BMI was selected and what criteria was used in deciding this.

Line 140: Please could you add few more details about assessment of PA.

Line 146: Please could authors be more specific – which groups were compared?

Please could the authors be more specific about how were the associations between diet patterns and variables of interest evaluated?

Results

Two factors were identified in principal component analysis. However, in practice it appeared that many participants consumed a mixture of traditional / westernized diets. How was this accommodated in the analyses?

I believe conventionally BMI > 25 is considered overweight – not obese (> 30).

Discussion

Line 302: “We speculate…” – this is somewhat confusing – why speculation? Authors have the data about the participants diet.

Line 346: “…is a useful dietary option for achieving weight loss…” – Not based on the results of this particular study.

As with the abstract, all the conclusions appear not to flow from the actual results / are over interpreted considering the study design.
